# Targeting the PD-1/PD-L1 Signaling Pathway for Cancer Therapy: Focus on Biomarkers

**DOI:** 10.3390/ijms26031235

**Published:** 2025-01-31

**Authors:** Areti Strati, Christos Adamopoulos, Ioannis Kotsantis, Amanda Psyrri, Evi Lianidou, Athanasios G. Papavassiliou

**Affiliations:** 1Analysis of Circulating Tumor Cells, Lab of Analytical Chemistry, Department of Chemistry, National and Kapodistrian University of Athens, 15771 Athens, Greece; lianidou@chem.uoa.gr; 2Department of Biological Chemistry, Medical School, National and Kapodistrian University of Athens, 11527 Athens, Greece; cadamop@med.uoa.gr (C.A.); papavas@med.uoa.gr (A.G.P.); 3Department of Oncological Sciences, Icahn School of Medicine at Mount Sinai, New York, NY 10029, USA; 4Department of Medical Oncology, Second Department of Internal Medicine, Attikon University General Hospital, Medical School, National and Kapodistrian University of Athens, 12462 Athens, Greece

**Keywords:** PD1/PD-L1, immune checkpoint inhibitor, immunological landscape, genomic landscape, transcriptomic landscape

## Abstract

The PD1/PD-L1 axis plays an important immunosuppressive role during the T-cell-mediated immune response, which is essential for the physiological homeostasis of the immune system. The biology of the immunological microenvironment is extremely complex and crucial for the development of treatment strategies for immunotherapy. Characterization of the immunological, genomic or transcriptomic landscape of cancer patients could allow discrimination between responders and non-responders to anti-PD-1/PD-L1 therapy. Immune checkpoint inhibitor (ICI) therapy has shown remarkable efficacy in a variety of malignancies in landmark trials and has fundamentally changed cancer therapy. Current research focuses on strategies to maximize patient selection for therapy, clarify mechanisms of resistance, improve existing biomarkers, including PD-L1 expression and tumor mutational burden (TMB), and discover new biomarkers. In this review, we focus on the function of the PD-1/PD-L1 signaling pathway and discuss the immunological, genomic, epigenetic and transcriptomic landscape in cancer patients receiving anti-PD-1/PD-L1 therapy. Finally, we provide an overview of the clinical trials testing the efficacy of antibodies against PD-1/PD-L1.

## 1. Introduction

Immuno-oncology is an evolving field of drug development that involves agents that harness the patient’s own immune system to fight cancer by counteracting the mechanisms by which the tumor evades the immune system [1]. Key effector cells, such as CD8+effector T cells, are components of the immune system that recognize and eliminate cancer cells through the recognition of tumor-specific antigens (neoantigens) [2]. The activation of T cells is regulated by immune checkpoints, which effectively limit and control an ongoing immune response to prevent autoimmunity. However, tumor cells (TCs) manage to evade the immune system by disrupting immune checkpoints. Overexpression of PD-1 ligands (programmed death-ligand 1, PD-L1) by TCs binds programmed cell death protein (PD-1) on T cells and inhibits their activity. Immune checkpoint inhibitors (ICIs) against PD-1/PD-L1 can release the brakes and lead to the activation of T cells [3].

The tumor microenvironment (TME) plays an essential role in tumor survival and function and mediates pro-tumorigenic effects through metabolic reprogramming [4]. The composition of the TME varies by tumor type, but characteristic features include cellular components and the extracellular matrix (ECM) that promote tumor growth, angiogenesis, invasion and metastasis [5]. M2-like tumor-associated macrophages (TAMs) [6] and cancer-associated fibroblasts (CAFs) [7] are essential components of the TME that facilitate tumor metastasis and enhance tumor drug resistance. Modulation of TAMs represents a novel therapeutic strategy and could be achieved by reducing or altering the function of TAMs [8]. Targeted therapies with CAFs include chemotherapy, immunotherapy and functional modification or reprogramming. However, many limitations such as the heterogeneity and plasticity of CAFs and the lack of specific target markers for CAFs may hinder the development of effective anti-CAF therapy [9].

Genomic landscapes have revealed great heterogeneity in different human cancers, as demonstrated by genome-wide sequencing studies [10]. However, the identification of somatic mutations and copy number alterations has been associated with potential response to immunotherapy [11]. Many independent studies have reported that tumor mutational burden (TMB) is a predictive biomarker for treatment success with ICIs in various tumor types [12,13,14]. On the other hand, cancer patients with low TMB may also have a good response to ICI monotherapies [15]. Different genomic landscapes have been found in tumors with different PD-L1 combined positive scores (CPS) [16], and it has also been reported that TMB and PD-L1 are independent biomarkers [17]. In this review, we focus on the function of the PD-1/PD-L1 signaling pathway and discuss the immunological, genomic, epigenetic and transcriptomic landscape in cancer patients receiving anti-PD-1/PD-L1 therapy. Finally, an overview of the clinical trials testing the efficacy of antibodies against PD-1/PD-L1 is provided.

## 2. Function of the PD-1/PD-L1 Signaling Pathway

### 2.1. T-Cell Response and Maintenance of Self-Tolerance

The PD1/PD-L1 axis plays an important immunosuppressive role during the T-cell-mediated immune response, which is essential for the physiological homeostasis of the immune system and the maintenance of self-tolerance [18]. T-cell activation is a multistep process that begins with the recognition of antigen by the T-cell receptor (TCR) bound to class I or II major histocompatibility molecules (MHCs) on the surface of antigen-presenting cells (APCs). This interaction forms a TCR–antigen–MHC complex that is supported by the CD4 or CD8 coreceptors [19]. Shortly after the activation of T cells, PD1 expression is induced on their surface, which prevents their overactivation and establishes an activation threshold [20]. T cells that cannot be further activated were initially termed “exhausted” [21]. This exhausted PD1-positive population has been shown to contain two subpopulations, both pre-exhausted and terminally exhausted cells, only the former of which can resume function after the PD1 blockade. This state of exhaustion is epigenetically determined, especially in terminally exhausted T cells [22]. It is evident that the downregulation of T-cell activation by PD1 and the subsequent suppression of the T-cell-mediated immune response has important physiological effects on the prevention of autoimmunity, the maintenance of self-tolerance and the maintenance of immune homeostasis [23]. Specifically, in regulatory T cells (Tregs), inhibitory PD-1 signaling, independent of Foxp3-mediated regulatory cell function, controls immune tolerance and the development of autoimmunity [24].

### 2.2. Effects of PD-1 and PD-L1 on Signal Transduction Pathways

Several signaling pathways have been reported to be affected by PD-1/PD-L1. Although there are still unresolved questions about which are the primary direct targets of PD1, between TCR and CD28, and the exclusivity of SHP2 in mediating PD1-derived inhibitory signaling, it is well established that PD1 targets multiple downstream signaling pathways [3,25]. These include members of the TCR signalosome ZAP70 and CD3zeta and downstream PKCtheta signaling [26], the Ras/mitogen-activated protein kinase (MAPK), phosphoinositide 3-kinase (PI3K)/Akt and mammalian target of rapamycin (mTOR) [27,28], including PTEN (phosphatase and tensin homolog) activation mediating PD1 [29].

### 2.3. PD1 Activation

The PD1 receptor is expressed on the surface of antigen-stimulated T and B cells and has two main ligands, PD-L1 and PD-L2, which have different expression patterns. Thus, PD-L1 is expressed in various cell types, including normal, hematopoietic and non-hematopoietic cells as well as cancer cells, whereas PD-L2 expression is mainly restricted to hematopoietic cells such as dendritic cells (DCs), macrophages, B cells, mast cells and activated T cells, but also tumor and stromal cells [20,30]. Although the expression of PD-L2 in head and neck tumors can be associated with an enhanced PD1-mediated T-cell response [30], PD-L2 is not considered a significant immunotherapy target in the clinic compared to PD-L1, the predominant inhibitory ligand of PD1 [20]. Activation of PD1 and subsequent inhibitory signaling occurs after binding of PD-L1 and PD-L2. The availability of PD-L1 can influence PD1 activation, as PD-L1 can also bind to the co-stimulatory molecule CD80. The binding affinity of PD-L1 to CD80 is slightly weaker than the binding affinity for PD1, whereas CD80 binds CD28 with a three-fold stronger binding affinity [31,32].

### 2.4. PD1 Expression

Apart from activated T cells, PD1 can be expressed on the surface of natural killer (NK) cells, B cells, macrophages and dendritic cells [33,34]. PD1 is encoded by the programmed cell death 1 (*PDCD1*) gene, whose transcription is induced by the TCR and NOTCH signaling pathways as well as various cytokines through the downstream activation of different transcription factors such as NFATc1, RBPJκ, STAT3/4/5, FOXO1, AP-1 and NF-κΒ [35]. PD1 expression can be epigenetically regulated via both DNA methylation and histone modification [36]. Thus, PD1 expression in naïve CD8+T cells is caused by hypermethylation of CRB and CR-C regulatory elements, whereas PD1 overexpression, which is associated with the phenotype of exhausted T cells, is induced by demethylation of the PD1 promoter [37,38]. In addition, PD1 expression can be regulated by post-translational modifications such as fucosylation and ubiquitination [39,40].

### 2.5. PD-L1 Expression

PD-L1 expression, reflecting a microenvironment rich in inflammatory cells, is considered a biomarker, albeit not a perfect one, for a favorable clinical outcome when administering ICIs [41]. PD-L1 expression can be regulated by gene amplification. For example, an increased copy number of the genetic locus encoding *CD274*, the gene responsible for PD-L1 protein synthesis, was found in patients with Hodgkin’s lymphoma and small-cell lung cancer (SCLC) [42,43]. Both extrinsic signals related to the immune response and intrinsic signals of the oncogenic pathway are involved in the transcriptional regulation of PD-L1 expression. For example, interferon gamma (IFNγ) produced by activated T cells and NK cells strongly triggers PD-L1 expression in the tumor microenvironment. Mutations in *IFNGR1/2* or *JAK1/2*, components of the IFNγ signaling cascade, are a common cause of acquired and primary resistance to ICI blockade therapies [44,45]. In addition, IFNγ targets such as PTPN2, APLNR and the chromatin remodeling complex PBAF have been identified by genetic screens as putative enhancers of ICIs in preclinical cancer models [46,47,48]. Apart from IFNγ, PD-L1 expression in various cancer cells can be induced by different immunostimulatory cytokines such as IL-6, INF-α and -β and TNF or immunosuppressive cytokines such as IL-10 and TGFβ [46].

### 2.6. Regulation of PD1/PD-L1 Expression in Cancer

PD-L1 expression is regulated in both cancer and immune cells by a variety of intrinsic stimuli, such as activation of the oncogenic EGFR, PI3K/AKT, MAPK and STAT3 signaling pathways, and extrinsic stimuli, such as various immunostimulatory cytokines [49]. PD-L1 expression can be regulated at post-transcriptional and post-translational levels. Regulation at the post-translational level involves modifications such as phosphorylation, glycosylation, ubiquitination and palmitoylation that affect PD-L1 protein stabilization [50,51], while post-transcriptional regulation is triggered by various oncogenic microRNAs that directly affect PD-L1 mRNA stability [52]. Recently, several studies have shed light on the direct or indirect effects of microRNAs (miRNAs) [53] and long non-coding RNAs (lncRNAs) [53,54] on the regulation of PD1/PD-L1 expression in different cancer types.

miRNAs bind directly to the 3’UTR of PD-L1 mRNA and lead to its degradation or suppression of translation. Overexpression of miR-4458 in NSCLC in vitro and in vivo models suppressed the inhibitory effect of the PD1/PD-L1 axis by targeting STAT3 [55]. In another study in human cisplatin-resistant lung cancer samples, miR-526b-3p was found to enhance the antitumor response via the STAT3/PD-L1 pathway [56]. In addition, PD-L1 expression in NSCLC was shown to be regulated by miR-34 in a p53-dependent manner [57]. Moreover, PD-L1 in lung cancer is a direct target of miR-200, whose transcriptional repression enhances the immunosuppressive effect of PD-L1 and metastatic potential [58]. In breast cancer (BC), miR-561-3p has been shown to downregulate PD-L1 expression by repressing certain oncogenes associated with breast cancer such as *ZEB1*, *HIF1A* and *MYC* [59].

lncRNAs can act as endogenous competitors of miRNAs and reverse their effects. In one such case, the lncRNA SNHG15 upregulates PD-L1 expression in gastric cancer by inhibiting miR-141 [60]. The lncRNA IncNDEPD1 can upregulate PD1 expression in cytotoxic T cells by directly interacting with miR-3619-5p and by blocking PDCD1 mRNA degradation [61]. Regulation of the PD1/PD-L1 axis in advanced melanoma patients treated with anti-PD1 therapy was demonstrated in a study that identified 15 lncRNAs that predicted survival benefit [62]. Recently, the association of the lncRNA NEAT with a favorable response to PD1/PD-L1 therapy in melanoma and glioblastoma patients was uncovered [63]. In gliomas, a group of six PD-L1-associated lncRNAs were identified to have prognostic value [64]. In BC, the upregulation of PD-L1 expression is caused by the inhibition of DNMT1-mediated methylation of miR-199a-5p induced by the lncRNA TINCR [65]. In addition, in BC patients, the lncRNAs XIST and TSIX were positively correlated with the expression of PD-L1 [66], and the lncRNA TCL6 correlated with the expression of PD1, PD-L1 and PD-L2 [67].

## 3. Landscape of Anti-PD-1/PD-L1 Therapy

### 3.1. Immune Landscape

The biology of the immunological microenvironment is highly complex and plays a crucial role in developing treatment strategies for immunotherapy (Figure 1) [68]. Characterizing this microenvironment and understanding the interactions between stromal and tumor cells is essential for identifying novel neoantigens [69]. Additionally, the definition of a tumor’s immunological subtype is key to predicting disease progression, often more so than the specific characteristics of individual cancer types. Identifying these neoantigenic immune targets can help elicit a durable immune response [68].

Characterization of the immune landscape of patients with metastatic melanoma has shown that it is possible to differentiate between responders and non-responders to anti-PD-1 monotherapy [70]. More specifically, in the responder group, high PD-L1 expression was found in macrophages at the tumor–stroma interface, followed by a rapid decrease in expression within the tumor, with even higher PD-L1 expression in the vicinity of cytotoxic T cells, whereas in non-responders, PD-L1 expression and cytotoxic T-cell activation follow a more arbitrary spatial regulation [70]. Tumors can be classified into “hot”, “intermediate” and “cold” groups according to their different immune profiles, which is highly consistent with the corresponding CD8+T-cell infiltration status [71]. Effective anti-PD-1 immunotherapy has also been associated with the presence of polyclonal CD8+T cells in the tumor with a limited number of mutations with lower TCR polyclonality [72]. CD8+T cells are characterized by a highly expressed CD161, while its ligand LLT1 (CLEC2D) is expressed in TCs, so the LLT1/CD161 distribution pattern is characterized by a high proportion of LLT1^+^ TCs and a low proportion of CD161^+^ CD8+T cells and is associated with a higher risk of lymph node metastasis [73]. In MSI-H/mismatch repair-deficient (dMMR) metastatic colorectal cancer (mCRC), the accumulation of more CD8+T cells in patients receiving first-line anti-PD-1 monotherapy is related to the sensitivity of the therapy [74]. On the other hand, the attenuation of CD8+T-cell function and the increase in immunosuppressive myeloid-derived suppressor cells (MDSCs) in metabolic dysfunction-associated steatohepatitis-related hepatocellular carcinoma (MASH-HCC) could be caused by squalene epoxidase (SQLE), which promotes an impaired antitumor response to anti-PD-1 therapy [75]. In addition, TREM2+ TAMs play an important role in suppressing CD8+T cells, while their deficiency inhibits tumor growth in vivo in HCC models by increasing the antitumor activity of CD8+T cells [76]. Moreover, in resistant pancreatic tumors in orthotopic pancreatic cancer (PaC) mouse models, the simultaneous presence of more depleted effector CD8+T cells and increased M2-like TAMs with a reduced capacity for antigen presentation has been reported [77].

Treg cells suppress anticancer immunity and contribute to immune evasion, tumor progression, and metastasis [78]. In experiments performed on mouse models of melanoma to analyze the intratumoral immune microenvironment, mice that did not respond to anti-PD1 therapy had a significantly higher population of Tregs that shifted the CD8+/Treg ratio in an age-dependent manner. Depletion of Tregs in melanoma using anti-CD25 was shown to reverse the response to anti-PD1 in non-responders [79], while in mice bearing a mammary tumor, neoadjuvant immune checkpoint blockade (ICB) was shown to remodel the intratumoral immune landscape characterized by an increase in CD8+T cells and NK cells and sustained T-cell activation [80]. The spatial distribution of T cells in the TME can predict the response to ICIs [81]. In renal cell carcinoma, there is a potential interaction between CD8+CD39+PD-1+ T cells and Foxp3+PD-1+ Treg cells due to the proximity between the cells, forming a spatial niche that is more specialized for immunosuppression under PD-1 blockade, which is more evident in metastatic lesions [82].

Computational algorithms have also revealed important information about tumor immune complexes in different cancer types [83]. In particular, immune cell infiltration clusters and gene clusters have been associated with different immune subtypes and survival outcomes in different cancer types [84,85]. In melanoma, a high immune cell infiltration score indicated activated immune properties and a better prognosis and was enriched with immune pathways and highly expressed immune signature genes [84]. Immune cell infiltration score was evaluated in HCC. It was found that in the high immune cell infiltration score group, activation of the Wnt/β-catenin pathway was significantly enriched and expression of immune checkpoint genes was increased, while the low immune cell infiltration score group was characterized by increased TMB and enrichment of metabolic-related pathways [86]. In lung adenocarcinoma (LUAD), a prognostic signature of the tumor microenvironment (LATPS) consisting of UBE2T, KRT6A, IRX2 and CD3D has shown that low LATPS is characterized by an immune phenotype, including an increased amount of ICI, tumor immune system dysfunction and increased activity of immune-related pathways, leading to better benefit from immunotherapy [87]. Similarly, in cervical cancer, an immune-related gene prognostic index (IRGPI) of the tumor immunological microenvironment (TIME) was able to predict the composition of immune cell subtypes such as macrophages, CD8 T cells and CD4 T cells, TIL exclusion and T-cell dysfunction [88]. In colon cancer, high IRGPI levels have been associated with cell adhesion molecules (CAMs) and chemokine signaling pathways, high macrophage M1 infiltration, suppressed immunity, a more aggressive colon cancer phenotype and lower therapeutic benefit of ICI treatment [89]. An NAD+ metabolism-related gene signature (NMRGS) has been used to establish a comprehensive risk model for glioma patients associated with a more immunosuppressive microenvironment and better therapeutic response to ICI therapy [90]. In head and neck squamous cell carcinoma (HNSCC), a novel immune signature (IMS) derived from a combined cohort identified hub genes and gene signatures for activated CD8+T cells correlated with the TIME that can predict response to immunotherapy, prognosis, immune infiltration and clinical features [91].

In non-small-cell lung cancer (NSCLC), a cold tumor immunological microenvironment, i.e., lower infiltration of CD3+ T cells and lower concentration of various immune signatures, correlated with a high chromosomal copy number variant (CNV) burden was associated with a lower benefit of ICIs, whereas a greater proximity of CD8+GZB+ T cells to malignant cells influenced the benefit of ICI therapy [92]. In addition, the macrophage landscape plays an important role in the prognosis of NSCLC patients. Patients with low infiltration of M2 macrophages have better overall (OS) and disease-free survival (DFS) [93]. B-cell infiltration is also an important factor in the sustained benefit of PD-1-based immunotherapy, as demonstrated in biopsy samples from HNSCC prior to treatment [94]. In addition, B-cell concentration in blood mononuclear cells (PBMCs) correlated strongly with tumor B-cell concentration in the tumor and showed a high predictive value for response to ICB [95]. A longitudinal study of the immunologic landscape in peripheral blood samples from NSCLC patients showed the association of immune cells with their correlating cytokines. CD8+and CD8+CD101hiTIM3+ (CCT T) were detected in low proportions of circulating cells from responders to immune ICB therapy, and CCT T cells secreted higher levels of cytokines [96]. Following ICI administration CD8+central memory T cells accumulate, and tumor-infiltrating lymphocytes (TILs) develop an effector-like phenotype over time [97]. In the MC38 tumor model, the TIL landscape changes in response to checkpoint blockade, as a switch from an NKT-driven TNFα response to a T-cell-driven IFN-γ response has occurred in the tumor microenvironment [98].

Fibroblasts may also play an important role in progression and the immune landscape, as has been reported for pancreatic ductal adenocarcinoma (PDAC). The functional heterogeneity of CAFs in PDAC differentially regulates cancer-associated signaling pathways and the accumulation of regulatory T cells [99]. Infiltration with inflammatory CAFs is also important for the immune mesenchymal-like (IML) subtype, which is associated with ECM activities and immune responses [100]. CAFs and macrophages may also contribute most to CD8 T-cell exclusion and immunosuppression by indicating activin A-mediated transcriptional reprogramming toward extracellular matrix remodeling, which polarizes the TME toward immunosuppression [101].

### 3.2. Genomic Landscape

The identification of potential genetic markers that could be associated with the benefit of immunotherapy is very important for the treatment of cancer patients receiving immunotherapy (Figure 1) [11,102]. However, genomic instability could be a barrier to improved sensitivity to ICIs [103]. The integration of high-throughput technologies such as next-generation sequencing (NGS), data extraction and analysis using the Cancer Genome Atlas (TCGA) database has enabled the prediction of genomic data from numerous samples of different cancer types [11] or genetic predispositions related to the immune system [104]. However, the heterogeneity of mutations in tumors can pose a major challenge for personalized medicine and biomarker development [105].

TMB has been postulated as a general determinant of ICI-dependent tumor rejection that depends on tissue and treatment context. A specific association between TMB and improved treatment outcomes with anti-PD-1/L1 therapies has been demonstrated [12,13,106]. Interaction effects between co-occurring genomic alterations on the efficacy of ICIs in NSCLC have been reported [107]. Large genomic data from 1846 patients with NSCLC showed that 44% of tumors had a targetable oncogenic alteration, with epidermal growth factor receptor (*EGFR*) being the most common. An exploratory analysis showed that the impact of TMB on survival with ICI treatment was associated with improved OS [108]. In addition, relatively high levels of TMB or DNA damage repair (DDR) alterations were found in esophageal squamous cell carcinoma (ESCC) patients prior to administration of neoadjuvant chemoimmunotherapy, suggesting that these different types of alterations are associated with therapeutic response [109]. Moreover, deficit mutations in checkpoint kinase 2 (*CHEK2*), an important gene in the DDR, enhance the antitumor effect of anti-PD-1 therapy, as shown in mice with MC38 and B16 tumors, by affecting the tumor immunological microenvironment [110].

Ataxia–telangiectasia mutations (ATMs) may also define a specific subset of NSCLC associated with Kirsten rat sarcoma virus (*KRAS*) mutations, increased TMB, decreased tumor protein P53 (TP53) and EGFR coexistence, and potentially increased sensitivity to combined therapy of ICIs and chemotherapy [111]. Transcription activator *BRG1* (*SMARCA4*)-mutated tumors are also very common recurrent alterations in NSCLC and are characterized by higher TMB and low or negative PD-L1, while the use of ICIs is associated with significantly improved survival [112]. In addition, PD-L1 expression and TMB have been shown to be independent biomarkers related to response rate to PD-1/PD-L1 inhibitors and could be generally used to categorize the immunological subtypes of cancer [17].

An analysis of intra-patient stability of TMB from tissue biopsies at different time points in advanced cancers has shown that TMB remains stable between tumor biopsies regardless of the time interval between sampling, but certain tumor types such as BC, colorectal carcinoma and glioma may experience an increase in TMB over time [113]. The landscape of frameshift (FS) mutations may also be important as a predictive biomarker for ICIs in combination with TMB, especially in tumors with low TMB, as FSs have been found in a high proportion of patients with low TMB (<10 mut/Mb) [15]. Similarly, patients with low TMB might benefit from ICIs when T-cell immunity is already present, without the need to distinguish between high- and low-TMB tumors [114].

The complexity of genomic analysis emphasizes the discovery potential of integrative analysis in large, well-curated, cancer-specific cohorts [115]. A pan-cancer study of 283,050 patient samples from multiple tumor types found that *CD274* (*PD-L1*) gene rearrangements may be important for ICI-prone cancers when other genomic alterations are present [116]. In addition, PD-L1-positive tumors (score ≥ 1) are associated with significantly higher exonic TMB, leading to enrichment of the PD-L1 pathway and other immune-responsive pathways [117]. Recently, an analysis of a cohort of 2504 patients from a broad spectrum of cancer types presented a new tumor classification system that includes 11 mutation-based genes and has the potential to support ICI treatment decisions [14].

On the other hand, the evolving mutational landscape may be related to acquired resistance to immune checkpoint blockade. Some of these mutations encode tumor neoantigens that can be recognized by T cells and could be exploited for the development of patient-specific immunotherapy approaches [118]. NSCLC patients with acquired resistance to EGFR-TKIs (tyrosine kinase inhibitors) could achieve superior efficacy of anti-PD-1/PD-L1 blockers, as specific T-cell responses could be achieved with high-value neoantigens generated from *EGFR* driver mutations [119]. It has been shown that a better understanding of the presentation of neoantigens by the tumor could be achieved if human leukocyte antigen-I loss of heterozygosity (HLA-I LOH) has the potential to refine TMB as a biomarker for response to checkpoint inhibitors [120]. Furthermore, PTEN loss in NSCLC tumors could lead to the development of an immunosuppressive microenvironment resulting in resistance to anti-PD-1 therapy, which can be overcome by targeting the immunosuppression mediated by PTEN loss [121].

### 3.3. Epigenetic Landscape

Epigenetic mechanisms underlie many aspects of antitumor immunity. The targeted use of epigenetic modifiers to remodel the immunological microenvironment holds great potential as an integral component of cancer treatment. Epigenetic reprogramming of immune cells underlies the development of tumor-reactive CD8+T cells and mediates the dysfunctional state of T cells and differentiation into MDSCs, while Treg cells possess a unique set of transcriptional and epigenetic features [122]. Understanding the epigenetic mechanisms in immune cells and their impact on immunologic responses in the TIME would enable the development of novel combinatorial treatments/biomarkers for correct treatment regimens and improved efficacy of immunotherapy (Figure 1) [123,124].

A risk model based on three m^5^C regulator-related genes in rectal adenocarcinoma (READ) patients was established and showed that high-risk patients had enrichment of cancer markers and were less sensitive to immunotherapy [125]. In addition, an m^6^A-related scoring system was developed to quantify the m^6^A modification pattern of individual samples using multiple datasets. Unsupervised clustering of 56 m^6^A regulator-related genes identified three distinct m^6^A regulator-related patterns, highlighting their important role in shaping TME diversity and clinical/biological features of gastric cancer [126].

METTL3 (Methyltransferase 3, N6-Adenosine-Methyltransferase Complex Catalytic Subunit) plays a key role in a variety of cancers, either dependent or independent of its m^6^A RNA methyltransferase activity [127]. Recently, M2-TAMs were shown to downregulate METTL3 in thyroid cancer cells and promote immunosuppressive TME through the transfer of extracellular vesicles (EVs), reversing the immunosuppressive effect of M2-TAMs that mediates resistance to anti-PD-1 therapy in thyroid cancer [128]. Following an unbiased clustered regularly interspaced short palindromic repeats–associated protein 9 (CRISPR–Cas9) epigenome screen, the tripartite motif containing 28-SET domain bifurcated histone lysine methyltransferase 1 (*TRIM28*-SETDB1) complex was identified as a regulator of PD-L1 expression. In a cohort of anti-PD-1 treated melanoma patients, *TRIM28* was significantly less expressed in responders compared to non-responders, while loss of *SEATB1* increased infiltration of effector CD8+T cells into the tumor microenvironment, a prerequisite for response to ICB [129].

### 3.4. Transcriptomic Landscape

Technologies such as RNA sequencing have identified new key factors and cellular subpopulations that create a microenvironment receptive to immunotherapy. The transcriptome at the single-cell level of innate and adaptive intratumoral immune cells has been studied in mouse models and in testing the efficacy of different types of chemoimmunotherapies [130]. Differentially expressed genes can reveal important biological signaling pathways in cancer patients undergoing anti-PD1 therapy (Figure 1) [131]. Transcriptome profiling can also reveal changes in gene expression during disease progression and clusters of genes based on relative expression changes between different stages [132].

An analysis of gene expression in lymphocytes within the tumor microenvironment co-expressing with the pan-T-cell marker CD3 epsilon subunit (*CD3E*) in 9601 human tumors from 31 cancer types was reported. Targets with a high *CD3E* correlation and a relatively low *PDCD1* correlation were identified, indicating the presence of T cells with low PD-1 expression and thus promising targets for therapy [133]. RNA-Seq data from TCGA data of patients with lung adenocarcinoma have revealed the gene landscape of PD-L1 and the status of B-cell infiltration, offering potential targets for future drug development [134]. Similarly, in melanoma cancer, six synthetic viability (SV) gene pairs were constructed as signatures to predict the clinical utility of ICIs [135].

Genetic mutations are also associated with RNA signatures, as demonstrated for myeloid inflammation in intrahepatic cholangiocarcinoma (iCCA). A negative feedback mechanism involves the upregulation of interleukin-1 receptor antagonist (IL1RN)-201/203 due to alternative splicing, which exerts important anti-inflammatory effects in *KRAS*-mutated iCCA that are significantly associated with a better response to anti-PD-1 immunotherapy [136]. Transmembrane protein 92 (*TMEM92*) mRNA expression has been identified as an immune resistance and prognostic marker in pancreatic cancer. More specifically, in patients with high expression of *TMEM92*, increased TMB is consistent with frequent mutations of *KRAS* and *TP53* [137].

Transcriptomic data have shown that angiogenesis and immune activities are exclusively enriched in the Alveolar Soft Part Sarcoma Chromosome Region, Candidate 1–Transcription factor E3 (ASPSCR1-TFE3) in renal cell carcinoma (rRCC), suggesting the superior clinical outcomes of ICIs and TKI combination therapy [138]. The stem cell landscape was also investigated in CRC using data from 1467 CRC samples. Three stem cell-related subtypes were characterized, with CRC patients with higher stem cell levels having a poorer prognosis, more immunosuppressive components in the TME and a lower response to immunotherapies [139].

## 4. Antibodies Targeting PD-1/PD-L1

ICIs have shown remarkable efficacy in a variety of malignancies in landmark trials and have fundamentally changed cancer therapy [140,141]. Over the past decade, several monoclonal antibodies targeting PD-1/PD-L1 have been shown to prolong survival and have been incorporated into routine clinical practice [1].

Although this novel approach of manipulating the immune system with monoclonal antibodies to generate an immune response was introduced by Chambers et al. in the 1990s [142], Hodi et al. were the first to publish the results of a landmark phase III trial showing a dramatic improvement in OS with the anti-Cytotoxic T-lymphocyte associated protein 4 (CTLA4) antibody ipilimumab in metastatic melanoma [143]. Since then, major melanoma treatment trials have shown a dramatic improvement in survival with anti-PD-1 inhibitors (Table 1) [144]. The final ten-year results of the CheckMate-067 trial were recently published [145]. CheckMate-067 is a landmark phase III trial that evaluated the combination of nivolumab and ipilimumab versus nivolumab monotherapy and ipilimumab monotherapy in metastatic melanoma. The median survival of patients receiving the combination was 71.9 months, compared with 36.9 months for nivolumab monotherapy and 19.9 months for ipilimumab monotherapy. Better survival outcomes were achieved with the combination therapy in patients with lower PD-L1 expression at the tumor level, while OS was similar between the two different groups of patients with PD-L1 expression in the tumor of 1% or more. As for the predictive value of *BRAF* mutations, these were associated with higher PFS and OS rates [145]. Since 2014, the anti-PD-1 antibodies pembrolizumab and nivolumab have been included in the melanoma treatment algorithm as standard treatment for all patients [146].

In addition, anti-PD-1/PD-L1 immunotherapy represents a major breakthrough in the treatment of NSCLC, as the analysis of PD-L1 expression has enabled the identification of patients with an increased likelihood of response to immunotherapy [154]. KEYNOTE-189 is a landmark phase III trial that evaluated the efficacy of pembrolizumab in combination with chemotherapy in metastatic NSCLC after a >1% PD-L1 tumor proportion score. More specifically the combination therapy resulted in a median OS of 22 months compared to 10.7 months with chemotherapy alone and was approved by the Food and Drug Administration (FDA) and the European Medicines Agency (EMA) as first-line treatment for metastatic NSCLC without actionable oncogenic alterations (AGAs) [147]. On the other hand, the IMpower150 study investigated the efficacy of the monoclonal anti-PD-L1 antibody atezolizumab in conjunction with the monoclonal anti-vascular endothelial growth factor (VEGF) antibody bevacizumab and chemotherapy. With a hazard ratio of 0.78, the combination treatment led to a median OS time of 19.2 months compared to 14.7 months in the control group (Table 2). Secondary and exploratory analyses showed that PFS was longer in the subgroup of patients who received atezolizumab with low or negative PD-L1 expression or high PD-L1 expression [155]. In locally advanced stage III NSCLC, the PACIFIC study investigated the monoclonal antibody durvalumab as a consolidation treatment after chemoradiation. Durvalumab showed a remarkable improvement in progression-free survival (PFS) (17.2 months compared to 5.6 months) and OS and has established itself as standard treatment in patients who respond to chemotherapy or have stable disease. The PFS benefit of durvalumab was observed independently of PD-L1 expression prior to chemoradiotherapy at a PD-L1 expression level of <25% and 0.41 at a PD-L1 expression level of ≥25% [156].

For recurrent or metastatic (R/M) HNSCC, pembrolizumab and nivolumab were established as first- and second-line treatments, based on the results of the KEYNOTE 048 and CheckMate 141 trials. KEYNOTE-048 showed that pembrolizumab was superior to chemotherapy in participants with untreated recurrent or metastatic HNSCC in patients with a PD-L1 CPS of 20 or more and a CPS of 1 or more populations [148]. On the other hand, the CheckMate-141 trial showed that nivolumab prolonged survival compared to standard chemotherapy or cetuximab in patients with platinum-resistant R/M disease. In the subgroup of patients with PD-L1 expression of 1% or more, treatment with nivolumab had a greater effect on OS therapy than in patients with PD-L1 expression of less than 1% [149]. These results have established ICIs as a fundamental component of the therapeutic algorithm for head and neck cancer [161].

In gastrointestinal cancer, predictive biomarkers such as MSI, PD-L1 expression, mutations in *KRAS* and *B-Raf* proto-oncogene, serine/threonine kinase (*BRAF*) genes, and the expression of receptor tyrosine-protein kinase erbB-2 (HER2) are crucial for treatment selection in first-line treatment. In MSI-H/dMMR mCRC, pembrolizumab has established itself as a first-line treatment after the results of the Keynote-177 trial showed the superiority of pembrolizumab compared to chemotherapy [150]. In gastric cancer, the Keynote-811 trial showed that pembrolizumab in combination with trastuzumab and chemotherapy improved OS in HER2-positive patients, with the greatest benefit observed in patients whose tumors expressed PD-L1 (CPS ≥1) [162]. In patients with advanced gastric or esophageal cancer, the CheckMate-649 trial showed that nivolumab in combination with chemotherapy provided a significant survival benefit. In addition, Nivolumab plus chemotherapy also provided better PFS in patients with a PD-L1 CPS of five or more, with a 32% reduction in the risk of progression or death compared to chemotherapy alone [151]. In addition, immunotherapy has shown great success in the treatment of liver cancer, especially HCC and cholangiocarcinoma. In patients with advanced HCC, the Imbrave 050 trial found a statistically significant difference in OS compared to sorafenib (20.2 months vs. 15.5 months) targeting both angiogenesis and PD-L1 signaling [160]. In addition, the HIMALAYA study investigated durvalumab in combination with a single dose of tremelimumab and durvalumab as monotherapy in comparison with sorafenib. This showed an improvement in the survival rate: 25% of patients were still alive after 4 years with the combination, compared to 15% with sorafenib [157]. However, recent data show that the tumor microenvironment in HCC is highly immunosuppressive, leading to new treatment approaches for HCC such as the use of ICIs as single agents or in combination with kinase inhibitors, anti-angiogenic drugs, chemotherapeutics and locoregional therapies [163]. A meta-analysis of second-line treatments for HCC has been reported highlighting the use of regorafenib and cabozantinib as established options for second-line treatments in the overall population [164]. For patients with advanced cholangiocarcinoma, TOPAZ-1 was the first study to show a survival benefit from the addition of durvalumab to standard chemotherapy [158].

RCC is considered a traditionally immunogenic tumor that was treated several years ago with first-generation immunotherapy (cytokines, interleukins, interferon). In intermediate- and high-risk patients, the CheckMate 214 trial showed that nivolumab/ipilimumab significantly improved OS and PFS compared to sunitinib. Specifically, longer OS and higher objective response rates were observed in intermediate- and low-risk patients across all PD-L1 expression levels, although the magnitude of benefit was higher in the population with PD-L1 expression of 1% or greater [165]. On the other hand, the combination of pembrolizumab and axitinib resulted in a median PFS of 15.7 months and an objective response rate of 60% [152]. In addition, the CheckMate 9ER study showed a significant improvement in survival in patients receiving nivolumab–cabozantinib versus sunitinib [166]. In addition, pembrolizumab was included in the treatment algorithm for patients with early-stage disease. The Keynote-564 trial demonstrated an extension of DFS at 24 months in high-risk renal cell carcinoma after nephrectomy [167].

Recent important research on advanced urothelial carcinoma has shown improved outcomes in initial treatment. Three important trials—CheckMate 901, EV-302 and JAVELIN—have significantly improved the treatment of bladder cancer. The CheckMate 901 trial showed that nivolumab in combination with chemotherapy led to a median OS of 21.7 months. The hazard ratios for OS and PFS were better in the nivolumab combination group than in the gemcitabine–cisplatin group, regardless of the PD-L1 expression level of patients’ tumors [168]. Most importantly, the combination of enfortumab vedotin and pembrolizumab resulted in an excellent OS of 31.5 months and a PFS of 12.5 months (EV-302/Keynote-A39 study) [153]. Maintenance immunotherapy with avelumab in patients without disease progression after platinum-based chemotherapy resulted in a significant benefit in OS and PFS in the JAVELIN 100 trial [159].

In the past, advanced gynecologic malignancies, including uterine and cervical cancer, were treated with conventional chemotherapy. The new molecular classification of endometrial cancer enables personalized therapy. Based on the Keynote-775 and GARNET trials, immunotherapy—including pembrolizumab and lenvatinib—has improved survival and PFS in patients with advanced endometrial cancer [169].

Immunotherapy has achieved encouraging results in the treatment of triple-negative breast cancer (TNBC). In particular, in patients with high PD-L1 expression, the KEYNOTE-355 study showed that pembrolizumab in combination with chemotherapy significantly extended survival. In patients with a PD-L1 CPS ≥10, the combination of pembrolizumab plus chemotherapy resulted in a median PFS of 9.7 months, compared to 5.6 months for chemotherapy alone [170].

## 5. Future Perspectives and Conclusions

The PD1/PD-L1 axis acts as a brake on the immune response and prevents overactivity, while T cells are the effector actors of the immune system that attack and destroy damaged cells such as virus-infected and cancer cells [171]. Expression of PD-L1 by tumor cells often activates PD1 signaling and allows the tumor to evade immune surveillance. Novel ICB and combination therapies for blocking the PD-L1/PD-1 pathway are leading to improved antitumor efficacy in patients with advanced cancers [1]. Characterizing the molecular landscape of the immunotherapy response by delineating activated cell states or cellular interactions may elucidate the molecular mechanisms of tumor response [70,96]. Even in patients who initially respond effectively to immunotherapy, the disease progresses over a period of 5 years. Resistance to immunotherapy is an evolving area of research that is challenging due to the complex and dynamic interplay between TCs and the immune system [172,173]. Key mechanisms of resistance include low TMB and heterogeneity of antigen presentation, alterations in the tumor microenvironment and host-related factors [174]. Current research is focused on strategies to maximize patient selection for therapy, clarify mechanisms of resistance, improve existing biomarkers, including PD-L1 expression and TMB, and discover new biomarkers [102,114].

In summary, anti-PD-1 and anti-PD-L1 antibodies have led to prolonged response and improved survival rates in a variety of malignant tumors. Ongoing research and clinical developments have established immuno-therapeutics as indispensable components of modern oncology.

## Figures and Tables

**Figure 1 ijms-26-01235-f001:**
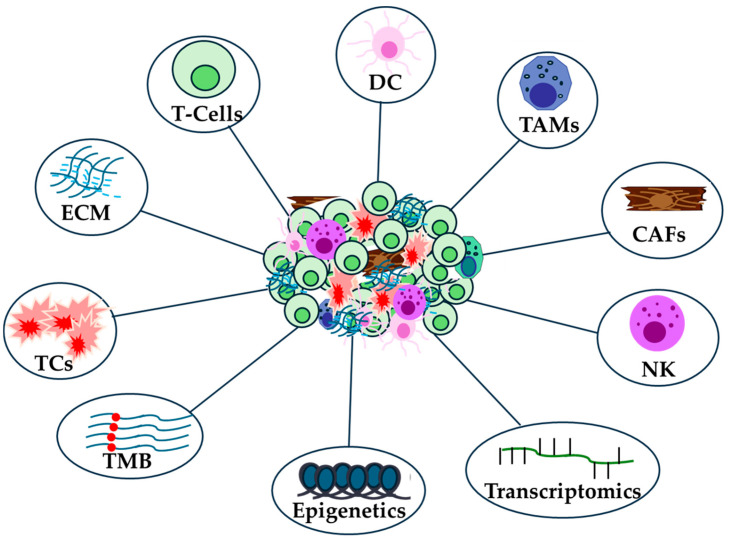
Immunological, genomic, epigenetic and transcriptomic landscape in cancer patients receiving anti-PD-1/PD-L1 therapy.

**Table 1 ijms-26-01235-t001:** Anti PD-1 inhibitors.

Clinical Trial (Phase III)	Drug Name (PD-1)	Cancer Type	Primary End Points	Response Rate (%)	Sample Size	FDA Approval (Yes/No)	Reference
CheckMate-067	Nivolumab + Ipilimumab vs. Nivolumab vs. Ipilimumab	Melanoma	PFS, OS	58	945	Yes	[145]
Keynote-189	Pembrolizumab + Chemotherapy vs. Chemotherapy + Placebo	NSCLC (Non-squamous)	OS, PFS	48.3	616	Yes	[147]
Keynote-048	Pembrolizumab + Chemotherapy vs. Pembrolizumab vs. Chemotherapy + Cetuximab	Head and Neck Squamous Cell Carcinoma (HNSCC)	PFS, OS	16.9	882	Yes	[148]
CheckMate-141	Nivolumab vs. Chemotherapy	HNSCC	OS	13.3	361	Yes	[149]
Keynote-177	Pembrolizumab vs. Chemotherapy + Bevacizumab or cetuximab	Colorectal Cancer (MSI-H/dMMR)	PFS, OS	43.8	307	Yes	[150]
CheckMate-649	Nivolumab + Chemotherapy vs. Chemotherapy	Gastric/Esophageal Cancer	OS, PFS	47.4	1581	Yes	[151]
CheckMate-214	Nivolumab + Ipilimumab vs. Sunitinib	Renal Cell Carcinoma	OS, PFS, ORR	42	1096	Yes	[152]
EV-302	Pembrolizumab + Enfortumab vedotin vs. Chemotherapy	Urothelial Cancer 1st Line	PFS, OS	44	608	Yes	[153]

**Table 2 ijms-26-01235-t002:** Anti PD-L1 inhibitors.

Clinical Trial (Phase III)	Drug Name (PD-L1)	Cancer Type	Primary End Points	Response Rate (%)	Sample Size	FDA Approval (Yes/No)	Reference
IMpower150	Atezolizumab + Bevacizumab + Chemotherapy vs. Bevacizumab + Chemotherapy	NSCLC 1st Line	PFS, OS	63	1202	Yes	[155]
PACIFIC	Durvalumab vs. Placebo after Concurrent Chemoradiotherapy	NSCLC (Stage III)	OS, PFS	28.4	713	Yes	[156]
HIMALAYA	Durvalumab + Tremelimumab (STRIDE) vs. Durvalumab vs. Sorafenib	Hepatocellular Carcinoma (HCC)	OS for STRIDE vs. Sorafenib	20.1	1171	Yes	[157]
TOPAZ-1	Durvalumab + Chemotherapy vs. Chemotherapy + Placebo	Biliary Tract Cancer	OS	26.7	685	Yes	[158]
JAVELIN 100	Avelumab vs. Best Supportive Care as Maintenance Therapy after 4–6 cycles Chemotherapy	Urothelial Cancer	OS	53.2	700	Yes	[159]
IMbrave050	Atezolizumab + Bevacizumab vs. Active Surveillance	Hepatocellular Carcinoma (HCC)		33	662	Yes	[160]

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
