# Peer review of "Targeting the PD-1/PD-L1 Signaling Pathway for Cancer Therapy: Focus on Biomarkers"

_ijms, 2025, doi:10.3390/ijms26031235_

Round 1

Reviewer 1 Report

Comments and Suggestions for Authors

Discrimination between potential responders and non-responders to anti-PD-1/PD-L1 therapy is an important  hot problem in cancer immunotherapy optimization. There is a number of recent reviews  about the role of the PD-1/PD-1L signaling pathway in cancer progression and its therapeutic targeting available.  This review article mainly summarizes the current knowledge about the tumor microenvironment in patients receiving anti-PD-1/PD-1L  therapy from immunological, genetic and epigenetic points of views and, moreover, brings an overview of the important clinical trials. So it is interesting for experts in the field. However, there is a lack of critical evaluation of the summarized data, especially of their relationships, and the most critical parameters could be more emphasized.

Specific comments:

-          Can you present data of the predictive markers identified in listed clinical trials more in detail?

-          Table 1 title should probably be “Anti PD-1 inhibitors” instead of “Anti PD-1/PD-L1 inhibitors”, and the title of the column “Drug Name (PD-1/PD-L1)” renamed, too.

Reviewer 2 Report

Comments and Suggestions for Authors

The manuscript by Strati et al. examines the role of the PD1/PD-L1 axis in cancer immune escape and its huge effect on cancer therapy. The manuscript could be improved. English should be revised.

I suggest the following points need to be addressed:

-          the authors should better describe the mechanisms that regulate PD-1/PD-L1 expression including the effects of microRNAs and lncRNAs on PD-1/PD-L1 expression in different cancers. A table or a figure summarizing the different regulations at molecular, post-transcriptional and epigenetic levels should be added.

-          Regarding immunotherapy in HCC, the great expression of immune checkpoint molecules on tumor and immune cells represents one of the major mechanisms of HCC immune escape. Recently, immunotherapy based on the use of immune checkpoint inhibitors, as single agents or in combination with kinase inhibitors, anti-angiogenic drugs, chemotherapeutic agents, and locoregional therapies, offers great promise to overcome the immunosuppressive tumor microenvironment and to improve the prognosis of HCC patients. The authors should elaborate more on this issue. The papers PMID 34065489 and PMID 34146196 could help.

-          The authors should briefly discuss the inefficacy of the immune checkpoint blockade therapies in some patients related to the phenotype of immune cells. This is the case for most multiple myeloma patients. Clinical trials based on immune checkpoint inhibitors monotherapy have provided unsatisfactory results. A possible explanation is that myeloma-specific T cells have an anergic or senescent phenotype rather than an exhausted phenotype, a prerequisite for the success of immune checkpoint blockade therapies. Moreover, during the treatment of myeloma patients with immune checkpoint inhibitors, the down-regulation of PD-1/PD-L1 expression leads to a compensatory up-regulation of other immune checkpoints. This aspect is well examined in the papers PMID 33194767 and PMID 38017516.

-          In the introduction, at line 35, I suggest to change the statement “through the expression of tumor-specific antigens” to “through the recognition of tumor-specific antigens” referring to T cells.

-          In the section “Landscape of anti-PD-1/PD-L1 therapy”, at line 203, I think that the abbreviation ICI for immune cell infiltration could be confused with the abbreviation ICIs for immune checkpoint inhibitors, and thus it should be removed. 

Comments on the Quality of English Language

English should be revised.

Round 2

Reviewer 2 Report

Comments and Suggestions for Authors

The authors have improved the manuscript which is suitable for publication.